# Are Three Weeks of Oral Anticoagulation Sufficient for Safe Cardioversion in Atrial Fibrillation?

**DOI:** 10.3390/medicina57060554

**Published:** 2021-05-31

**Authors:** Stefan Naydenov, Nikolay Runev, Emil Manov

**Affiliations:** Department of Internal Diseases “Prof. St. Kirkovich”, Medical University of Sofia, 1431 Sofia, Bulgaria; nrunev@abv.bg (N.R.); doctor_emil_manov@abv.bg (E.M.)

**Keywords:** anticoagulant, atrial, thrombosis, dabigatran, monitoring

## Abstract

*Background and Objectives:* Patients with atrial fibrillation (AF), lasting >48 h, considered for cardioversion, are recommended ≥3 weeks of oral anticoagulation before sinus rhythm restoration because of high risk of development of left atrial thrombosis (LAT) and stroke. However, the optimal duration of anticoagulation in the presence of overt LAT is unknown. *Materials and Methods:* An open-label study aimed to investigate the prevalence of spontaneous echo contrast (SEC) and LAT before and after 3 weeks of direct oral anticoagulant (DOAC) treatment. We included 51 consecutive patients (50.9% males), mean age 69.3 ± 7.4 years with paroxysmal/unknown duration of AF, considered for cardioversion, who agreed to have transesophageal echocardiography at enrollment and 3 weeks later. *Results:* At baseline SEC was present in 26 (50.9%) and LAT in 10 (19.6%) of 51 patients. After 3 weeks on DOAC, SEC persisted in 12 (25.0%) and LAT in 7 (14.5%) of 48 patients, *p* < 0.05 vs. baseline. Factors, associated most strongly with persistence of SEC/LAT, were left atrial appendage (LAA) emptying velocity <20 cm/s (OR = 2.82), LAA lobes >2 (OR = 1.84), and indexed left atrial volume ≥34 mL/m^2^ (OR = 1.37). *Conclusions:* In our study the incidence of SEC/LAT, particularly in AF with unknown duration, was not as low as we expected. The prevalence of SEC/LAT seemed to be dependent on factors not routinely evaluated in AF patients planned for cardioversion (indexed LA volume, LAA morphology and number of lobules, LAA emptying velocity, etc.). Our data suggested an individualized approach for DOAC duration in AF patients before an attempt for restoration of sinus rhythm is made, taking into consideration the LAA morphology and function.

## 1. Introduction

Atrial fibrillation (AF) is the most common arrhythmia in humans, with estimated prevalence ~3% among the general population aged >20 years: One in four middle-aged adults will develop at least one episode of this rhythm disorder [1]. In addition, AF is a strong and independent risk factor (RF) for ischemic stroke and/or peripheral arterial thromboembolic events [1,2,3,4]. Even short/rare episodes of AF possess certain risk of stroke, which should not be neglected, and the risk increases significantly with AF duration, aging, and in the presence of other concomitant diseases such as arterial hypertension, heart failure, chronic kidney disease, diabetes, etc. [1,2,3,4].

Many patients with episodes of paroxysmal/persistent AF are referred for pharmacological/electrical cardioversion: Rhythm control is a part of the symptom control strategy (together with rate control) [1,2,3]. However, cardioversion (pharmacological or electrical) is associated with 5–7% risk of thromboembolic events within 30 days in non-anticoagulated patients [2,4,5]. Thromboembolic rates are highest within the first 7 days after cardioversion (>80% of events), with 70% of them occurring the first 2 days [2,5,6,7,8,9].

Clinical trials show that anticoagulation prior, at the time, and post-cardioversion can reduce the 30-day thromboembolic risk significantly (<1%) [2,5,10]. For this reason, the current European Society of Cardiology (ESC) guidelines recommend a minimal period of 3 weeks of oral anticoagulation (OAC) to patients with AF lasting ≥48 h or AF with unknown duration or AF duration 12–48 h with CHA2DS2-VASc (congestive heart failure, hypertension, age ≥75 years, diabetes mellitus, stroke/transient ischemic attack, vascular disease, age 65–74 years, and female gender) score ≥2 for males and ≥3 for females, if the patients are considered for cardioversion, but left atrial thrombosis (LAT) cannot safely be excluded by a transesophageal echocardiography (TEE) [1]. It has to be mentioned that these recommendations are not based on results from randomized clinical trials [1,2,11]. In addition, there are a lot of reports of peri-/post-cardioversion strokes even in well-coagulated (according to the guidelines) patients with low CHA2DS2-Vasc score [5,6,7]. Many of these events are due to LATs that have been formed before initiation of OAC and persisting after the recommended minimal period of anticoagulation [8,9,12].

Data about the real prevalence of LAT in patients with paroxysmal/persistent AF as well as the prevalence of spontaneous echo contrast (SEC), a very risky finding, predisposing to LAT formation, are still insufficient [7,8,9,10,11,12,13,14,15]. The traditional therapeutic approach to these patients is treatment with parenteral anticoagulants or vitamin K antagonists (VKAs), mostly warfarin [1,8,12]. The current ESC guidelines also suggest that the direct oral anticoagulants (DOACs) could be as efficient and safer as VKAs. However, robust data from randomized clinical trials (RCTs) are lacking [10,11,12,13,14,15,16,17,18,19].

Giving the importance of optimal OAC in AF patients considered for cardioversion we conducted a study, aiming to investigate the prevalence of SEC/LAT in patients with paroxysmal/unknown duration of AF, before DOAC treatment and 3 weeks later, thus evaluating the potential of DOACs to resolve these pathological conditions. 

## 2. Materials and Methods

This prospective, open-label study was conducted in the period of May 2016–October 2017. We included 51 consecutive out-hospital patients with paroxysmal, non-valvular AF, planned for pharmacological/electrical cardioversion (26 males—50.9%, *P* males/females = N.S.), mean age 69.3 ± 7.4 (61–86) years. All patients were examined with transesophageal echocardiography (TEE) at baseline, before DOAC initiation, and 3 weeks later. The conductance of our study was in accordance with the ethical standards of the Institutional or Regional Responsible Committee on human experimentation and with the Helsinki Declaration of 1975, as revised in 1983. A written informed consent was obtained from all patients before their enrolment in the study, which was approved by the local Ethics Committee. 

### 2.1. Patients’ Selection

The following eligibility criteria were considered for enrolment in the study.

Inclusion criteria were (1) age ≥ 18 years; (2) paroxysmal AF according to the classification of this arrhythmia in 2016 ESC Guidelines for diagnosis and treatment of AF (not changed in 2020 Guidelines) or AF with unknown duration, considered for rhythm control; (3) hemodynamic stability; (4) symptomatic patients in III–IV functional classes (severe or disabling symptoms of AF, normal daily activity affected or discontinued) according to the 2016 European Heart Rhythm Association (EHRA) classification; (5) rhythm control strategy (elective cardioversion) adopted by both the treating physician and the patient; (7) fulfilled criteria for DOAC treatment of AF patients according to 2016 ESC Guidelines; and (8) signed, written, informed consent by the patients. 

Exclusion criteria were: (1) AF other than paroxysmal/persistent and/or not considered for sinus rhythm restoration (pharmacological or/and electrical) irrespective of the reason; (2) patient’s disagreement to be treated with a DOAC or to be followed up for the study period; (3) documented prior stroke, transient ischemic attack, or peripheral embolic event; (4) planned elective procedure of any kind, requiring interruption of the DOAC during the study period; (5) DOAC treatment, initiated before inclusion in the study; and (6) mentally disabled patients, unable to understand the written, informed consent or any other patients unwilling to sign the informed consent for participation in the study for any reason.

All screened patients were prescribed a DOAC according to the 2016 ESC Guidelines (not changed in 2020 Guidelines). The choice of the type of a DOAC and its dosage was made by the treating physician based on patient’s clinical and instrumental findings, DOAC prescription rules, and individual preference. 

At baseline all participating patients had the following instrumental investigations: electrocardiography (ECG), transthoracic echocardiography (TTE), and routine laboratory parameters. A control TEE was performed 3 weeks later in 48 of the patients (three patients refused re-assessment).

TTE and TEE were done in a standard manner with GE Vivid E95 (GE Healthcare, Chicago, IL, USA). Before TEE, we applied light conscious sedation and topical anesthesia. Multiplane transducer was used for TEE imaging: Depending on the image quality, the frequency and gain settings were carefully adjusted to minimize gray-noise artifact, especially when assessing for SEC. The echocardiographic images were acquired using the standard procedure and digitally stored. A thrombus was defined as an echo-dense mass and SEC was considered present when dynamic “smoke-like” echoes were seen in the left atrium. In addition, the LA appendage flow velocity was also obtained by the pulsed Doppler method with the sample volume placed 1 cm distally from the opening of the appendage.

Patients’ information was collected in a structured questionnaire form (demographic characteristics, socio-economic data, history about clinical presentation, risk factors, comorbidities, and medical treatment). We gathered the necessary data directly from the patient’s medical history and/or from the available medical documentation. 

The design of our study is shown in Figure 1.

### 2.2. Statistical Analysis

The statistical analysis was performed by SPSS statistical package, version 16.0 (SPSS Inc., Chicago, IL, USA). The data were summarized by frequencies and percentages for categorical variables and by minimal, maximal, mean values, and standard deviation for continuous ones. For comparison of categorical variables, we used independent χ2-test and Fisher’s exact test. The normality of distribution of continuous data was assessed by Shapiro–Wilk test. T-test and ANOVA were used for comparison of parametric data and the Mann–Whitney U test (with Bonferroni correction when necessary) was used for non-parametric data. Univariate and multivariate logistic regression was performed to identify factors associated strongly with persistence of SEC/LAT after DOAC treatment, with odds ratio (OR) demonstrating the strength of each factor. All results were considered to be statistically significant at *p*-values < 0.05.

## 3. Results

The most important baseline characteristics of the enrolled patients are shown in Table 1. There were no statistically significant gender differences in terms of the age, body mass index, concomitant diseases, stroke and bleeding risk, and other analyzed variables in our study population.

Baseline TEE showed SEC in the left atrial appendage (LAA) of 26 out of 51 patients (50.9%) and a thrombus in another 10 (19.6%). In terms of grading, SEC was mild (grade 1+) in six (23.1%), mild to moderate (grade 2+) in eight (30.8%), moderate (grade 3+) in seven (26.9%), and severe (grade 4+) in five (19.2%).

Table 2 presents the baseline TEE findings of the patients with and without SEC/LAT, before initiation of a DOAC (*n* = 51).

A repeated TEE was performed on 48 patients. Three patients who had no SEC/LAT at baseline refused to have a second TEE. Among the patients with baseline SEC/LAT, after 3 weeks on DOAC treatment SEC persisted in 12 (46.2%) of 26 patients (*p* < 0.001 versus baseline SEC prevalence): It was mild in seven (58.3%), mild to moderate in four (33.3%), and moderate in one (8.3%). A thrombus was present in seven (70%) of 10 with baseline LAT (*p* = 0.024 versus baseline TEE). In four of these patients there was reduction of thrombus size and in the other three there were no visual changes after the DOAC treatment. Among the patients free of SEC/LAT at baseline TEE, there were no newly developed findings of this kind at the repeated echo investigation.

Table 3 presents the TEE findings at Week 3 of the patients who were diagnosed with SEC/LAT at baseline. All patients with resolution of these findings after 3 weeks of DOAC treatment were on dabigatran (all of them on 150 mg b.i.d).

Table 4 shows parameters that have been found to be significantly associated with the risk of non-reversal of SEC/LAA thrombus after 3 weeks of DOAC treatment (mostly dabigatran).

During the follow-up period no cerebro-vascular or peripheral arterial ischemic events were observed in our study population. No other new cardiac (myocardial infarction, decompensation of HF, etc.) or non-cardiac complications occurred either. The DOAC treatment was well tolerated with compliance and persistence >80%. 

## 4. Discussion

In our study we investigated the baseline and after-treatment prevalence of SEC/LAT in AF patients with non-valvular, paroxysmal, or of unknown duration AF scheduled for pharmacological or electrical cardioversion. We included symptomatic patients (high EHRA Score) with heart rate not well controlled by optimal pharmacological rate-control therapy (in accordance with the current ESC Guidelines), for which we considered rhythm-control strategy. The modern AF therapeutic strategy is based on three main pillars, the so called “ABC pathway”: “A” is to avoid stroke with anticoagulants, “B” is for better control of symptoms with patient-centered decisions on rate or rhythm control, and “C” is for cardiovascular and comorbidity risk optimization [20]. Because of troublesome AF-related symptoms (mainly due to the high ventricular response) impairing daily physical activities and quality of life, the recommended minimal 3-week period on OAC before cardioversion was an unacceptable option for our patients. So, we considered early cardioversion after TEE to exclude LAT/SEC. Transesophageal echocardiography is currently the modality of choice for evaluation of the LAA [9,21,22]. It allows complete delineation of the LAA anatomy, assessment of its function, and detection of LAT [9,22]. 

TEE confirmed SEC/LAT in 36 (70.6%) of our non-anticoagulated AF patients: 26 (51%) with SEC of different degree and 10 (19.6%) patients with LAA thrombus. In other AF studies LAT was observed in 5–30% [2,3,4,5]. In a study with 262 non-rheumatic AF patients, TEE revealed intracardiac thrombi in 8% [23]. Another study showed similar results: 8.8% of patients with non-valvular AF on a VKA with sub-therapeutic OAC therapy were diagnosed with LAT, whereas a thrombus was observed in 3.6% of patients with sufficient anticoagulation therapy [8]. The higher incidence of SEC/LAT among our AF patients could be explained with (1) the possible longer duration of AF episodes (>48 h) despite recognized paroxysmal type of the arrhythmia at the moment of diagnosis; (2) lack of prior/ongoing anticoagulant therapy; and (3) high-risk left atrial appendage morphology (especially number of lobes).

Some studies found that higher LAT was associated with higher CHA2DS2 -VASc score, prior SEC, older age, female gender, concomitant HTN, DM, and chronic HF [5,8]. 

Most of the published TEE data from AF patients focused on the presence of LAT but not on SEC, which is much more prevalent and probably also a strong predictor of stroke [1,2,6]. The real prevalence of SEC in AF patients is largely unknown. In our study it was found in every second patient considered for restoration of sinus rhythm. The factors that promote progression of SEC to LAT in different patients with similar clinical risk profile (including LAA morphology) remain unclear. In our study population, analysis did not disclose any factors/conditions to be significantly associated with increased risk of SEC progression to LAT.

We performed a control TEE after 3 weeks of DOAC treatment, aiming not only to evaluate the resolution rate of LAT SEC but also to disclose predictors associated with higher risk of non-resolution of these findings. In terms of OAC choice, it has to be mentioned that observational and prospective data had not shown significant difference between DOACs and VKAs in LAT resolution in AF patients [5,10,11,12,13,14,15]. Traditionally, VKA therapy was recommended as a standard of care with strict follow-up and INR monitoring until resolution of LAT (with heparin bridging if necessary) [1,2,3,4,5]. It is now generally admitted that DOACs may be an option for LAT resolution, where VKAs are not well tolerated or sufficient INR control cannot be obtained [1,10,11,12,13,14,15,16]. 

Among our patients with baseline SEC/thrombosis, after 3 weeks of DOAC treatment (mostly with dabigatran) SEC had been completely resolved in 14/26 (53.8%) but LAT in 3/10 (30%) only. X-TRA study indicated LAT resolution rate of 41.5% with rivaroxaban (20 mg o.d.), comparable to the retrospective CLOT-AF registry, in which LAT resolution was observed in 62.5% in heparin/warfarin-treated patients [24]. In the EMANATE trial, thrombus resolution rate was similar in patients treated with apixaban (52%) as with conventional therapy (56%, 10/18) [25]. Collins et al. performed TEE before and after 4 weeks on OAC in 18 patients with nonvalvular AF and reported that LAT was resolved in 16 (89%) of them [26]. All these data are quite important for the clinicians because in the real clinical practice TEE is not routinely performed in short-term AF (<48 h) or after sufficient OAC therapy (>3 weeks) [2,3,11]. In our study, predictors for LAT/SEC non-resolution were LAA emptying velocity <20 cm/s, associated most strongly with the risk of persistence of SEC/LAA, followed by indexed LA volume >40 mL/m^2^ and presence of >2 LAA lobules.

### Study Limitations

There were several limitations of our study. (1) The number of patients was relatively small, especially of patients with LAT. Therefore, our findings could not be extrapolated to the entire AF population treated with DOAC for the same period of time. (2) Most of our patients were treated with dabigatran and we could not make general conclusions for the other DOACs. (3) The 3-weeks DOAC treatment showed improvement of SEC and reduction of LAT size in some patients, although they persisted. If longer follow-up was performed, the number of patients free of LAT and particularly of SEC could be higher. (4) We investigated some, but not all, factors that might be related to the persistence of SEC/LAT after sufficient period of anticoagulation. 

## 5. Conclusions

In our study the incidence of SEC/LAT, particularly in AF with unknown duration, was not as low as we expected. The prevalence of SEC/LAT seemed to be dependent on factors not routinely evaluated in AF patients planned for cardioversion (indexed LA volume, LAA morphology and number of lobules, LAA emptying velocity, etc.). Our data suggest an individualized approach for DOAC duration in AF patients before an attempt for restoration of sinus rhythm is made, taking into consideration the LAA morphology and function. 

## Figures and Tables

**Figure 1 medicina-57-00554-f001:**
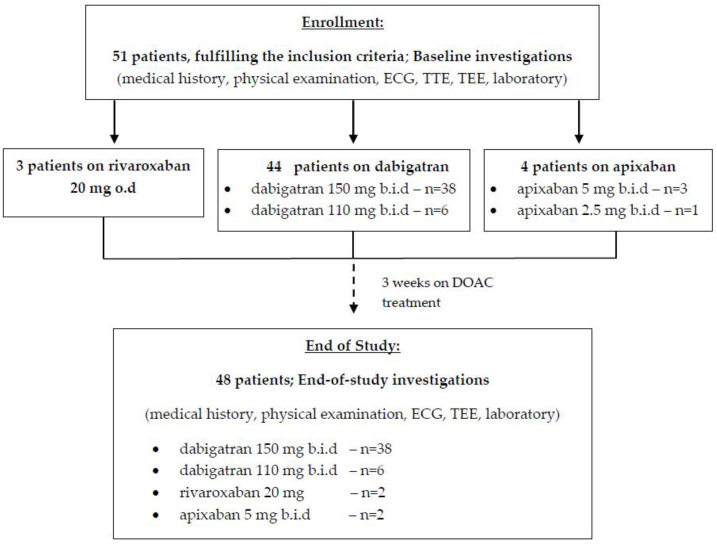
Study design: patient’s enrolment and follow-up schedule; ECG, electrocardiography; TEE, transesophageal echocardiography; TTE, transthoracic echocardiography.

**Table 1 medicina-57-00554-t001:** Baseline characteristic of the enrolled population.

Parameters	Total	Males	Females	*p*
	*n* = 51	*n* = 26	*n* = 25	
Age (years)	67.3 ± 7.4	66.4 ± 5.7	68.6 ± 6.1	0.126
Body mass index	28.8 ± 3.4	28.7 ± 5.2	29.5 ± 4.9	0.472
HTN, *n* (%)	46 (90.2%)	22 (84.6)	24 (96.0)	0.160
Type 2 DM, *n* (%)	10 (19.6)	4 (15.4)	6 (24.0)	0.048
Vascular disease *^,#^, *n* (%)	6 (11.8)	3 (11.5)	3 (12.0)	0.742
VHD ^&^, *n* (%)	2 (3.9%)	1 (3.9)	1 (4.0)	0.778
CHA2DS2-VASc (points)	3.92 ± 1.12	3.86 ± 1.04	4.17 ± 0.96	0.010
HAS-BLED (points)	2.33 ± 0.88	2.47 ± 1.07	2.26 ± 0.66	0.622
CrCl (ml/min.)	57.4 ± 7.02	54.1 ± 6.23	59.7 ± 8.16	0.276
Indexed LA volume (mL/m^2^)	31.8 ± 6.44	30.2 ± 5.04	32.6 ± 7.38	0.213
LVEF (%)	64.8 ± 6.46	66.2 ± 7.22	62.9 ± 8.17	0.381
dabigatran 150 b.i.d	38 (74.5)	18 (69.2)	20 (80.0)	0.340
dabigatran 110 b.i.d	6 (11.8)	4 (15.4)	2 (8.0)	0.220
apixaban 5 mg b.i.d	3 (5.9)	1 (3.8)	2 (8.0)	0.338
apixaban 2.5 mg b.i.d	1 (2.0)	0 (0)	1 (4)	0.191
rivaroxaban 20 o.d	3 (5.9)	3 (11.5)	0 (0)	0.083

CHA2DS2-VASc—stroke risk score in atrial fibrillation (C—chronic heart failure, H—hypertension, A—age ≥ 75 years, D—type 2 diabetes mellitus, S—suffered stroke or transient ischemic attack, VA—vascular disease, Sc—sex category); CrCl—creatinine clearance, calculated by the Cockroft-Gault original formula; DM—type 2 diabetes mellitus; HAS-BLED—bleeding risk score for patients with atrial fibrillation on oral anticoagulation (H- uncontrolled hypertension, A—abnormal renal or liver function, S—prior stroke, B—bleeding, L—labile international normalized ratio, E—elderly, aged ≥65 years, D—drugs or alcohol); HTN—arterial hypertension; LA—left atrial; LVEF—left ventricular ejection fraction, measured by transthoracic echocardiography; VHD—valvular heart disease; * documented coronary artery disease (including post-myocardial infarction), peripheral arterial disease, and/or aortic plaque; ^#^ concomitant atherosclerotic cerebro-vascular disease was an exclusion criterion; ^&^ any moderate to severe VHD except for moderate to severe mitral stenosis or a prosthetic valve (contraindications for DOACs).

**Table 2 medicina-57-00554-t002:** Baseline TEE results.

Parameters	SEC/LAT (−) *n* = 15	SEC/LAT (+) *n* = 36	*p*
Indexed LA volume (mL/m^2^)	27.8 ± 10.2	35.4 ± 8.6	<0.001
LAA filling velocity (cm/s)	43.4 ± 7.2	39.5 ± 10.6	0.218
LAA emptying velocity (cm/s)	29.4 ± 11.3	18.7 ± 14.4	0.033
Number of LAA lobes, *n* (%)			
1	10 (66.7)	5 (13.9)	<0.001
2	3 (20.0)	13 (36.1)	0.045
3	2 (13.3)	12 (33.3)	0.012
>3	0 (0)	6 (16.7)	<0.001

LA—left atrial; LAA—left atrial appendage; LAT—left atrial thrombosis; SEC—spontaneous echo contrast; TEE—transesophageal echocardiography.

**Table 3 medicina-57-00554-t003:** TEE parameters of patients who had baseline SEC/LAA thrombus after 3 weeks of DOAC treatment in terms of persistence of these findings (*n* = 48).

Parameters	SEC/LAA Thrombus * (−) *n* = 29	SEC/LAA Thrombus (+) *n* = 19	*p*
Indexed LA volume (mL/m^2^)	32.3 ± 6.2	37.4 ± 9.5	0.020
LAA filling velocity (cm/s)	40.4 ± 8.1	33.6 ± 9.5	0.014
LAA emptying velocity (cm/s)	24.8 ± 7.1	15.6 ± 9.2	<0.001
Number of LAA lobes, *n* (%)			
1	13 (44.8)	0 (0)	<0.001
2	12 (41.4)	4 (21.1)	<0.001
3	4 (13.8)	9 (47.4)	0.040
>3	0 (0)	6 (31.6)	<0.001

LA—left atrial; LAA—left atrial appendage; LAT—left atrial thrombosis; SEC—spontaneous echo contrast; TEE—transesophageal echocardiography; * three patients free of SEC/LAT at baseline refused to have a repeated TEE (two of these patients had unilobular LAA and one had three-lobular LAA according to the results from the baseline TEE).

**Table 4 medicina-57-00554-t004:** Association between some of the examined TEE parameters and the risk of non-resolution of SEC/LAT after DOAC treatment.

Variable	OR	95% Confidence Interval for OR		*p*
		Lower Limit	Upper Limit	
Indexed LA volume ≥34 mL/m^2^ vs. 34 mL/m^2^	1.37	1.28	1.88	0.036
LAA emptying velocity <20 cm/s vs. ≥20 cm/s	2.82	1.94	4.38	<0.001
Number of LAA lobes >2 vs. ≤2	1.84	1.42	2.93	0.042

LA—left atrial; LAA—left atrial appendage; LAT—left atrial thrombosis; SEC—spontaneous echo contrast; OR—odds ratio; TEE—transesophageal echocardiography; vs.—versus.

## Data Availability

All materials supporting reported results are available with Nikolay Runev, e-mail: nrunev@abv.bg, Department of Internal Diseases “Prof. St. Kirkovich”, Medical University of Sofia, Bulgaria.

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
