# Peer review of "Are Three Weeks of Oral Anticoagulation Sufficient for Safe Cardioversion in Atrial Fibrillation?"

_medicina, 2021, doi:10.3390/medicina57060554_

Round 1
Reviewer 1 Report
Oral anticoagulant (OAC) therapy with vitamin K antagonists (VKAs) and new oral anticoagulants (NOACs) is highly effective in the prevention of stroke for patients with atrial fibrillation (AF). The use of NOACs gave important benefits compared to VKAs in reducing the probability of major hemorrhagic events including intracranial bleeding (IB) with similar or better stroke prevention and easier management. The authors studied a group of patients with atrial fibrillation (AF), lasting >48 h, considered for cardioversion. For these patients are recommended ≥3 weeks of oral anticoagulation before sinus rhythm restoration because of high risk of development of left atrial thrombosis (LAT) and stroke.This is an open-label study, aiming to investigate the prevalence of spontaneous echo contrast (SEC) andLAT before and after 3 weeks of direct oral anticoagulant (DOAC) treatment. The authors included 51 consecutive patients (50.9% males), mean age 69.3±7.4 years with paroxysmal/unknown duration of AF, 14considered for cardioversion, who agreed to have transesophageal echocardiography at enrollment 15and 3 weeks later.
At baseline SEC was present in 26 (50.9%) and LAT in 10 (19.6%) of 51 16 patients. After 3 weeks on DOAC, SEC persisted in 12 (25.0%) and LAT in 7 (14.5%) of 48 patients, 17 p < 0.05 vs baseline. Factors, associated most strongly with persistence of SEC/LAT were left atrial 18 appendage (LAA) emptying velocity < 20 cm/s (OR = 2.82), LAA lobes >2 (OR = 1.84) and indexed 19 left atrial volume ≥ 34 ml/m2 (OR = 1.37).
The conclusions are that the recommended minimal period of 3 weeks 20 of oral anticoagulation lead to SEC/LAT resolution in 47% of our patients and LA/LAA 21 morphology and function should be taken into account when determining the duration of DOAC 22 treatment before planned cardioversion. The study is interesting but presents several critical issues. The major limitations are that the percentage of thrombosis and sludge is very high and the number of patients analyzed is too low to draw conclusions. The same goes for the morphology of the auricle which in some cases is probably at greater embolic risk but also in this case it is necessary to have a larger study population.
Author Response
Dear Reviewer,
Thank you for reviewing our manuscript! We highly appreciate your comments!
- To your comment about the small number of patients in our study:
We agree with you that the number of patients is not large enough to draw general conclusions. We have admitted this fact in “Study limitations”. When we searched the literature about published articles in this sphere, we found that most of the data about LAT treatment are derived actually from small studies, including similar number of patients like ours. And many were based only on single case-reports or summary of case reports. For example, one of the last published articles on this topic (Fleddermann A, et al. American Journal of Cardiology, January 2020) was based on 33 patients only.
We included in our study all patients we came upon during the study period – patients, meeting the inclusion criteria (and without the corresponding exclusion criteria). As our study ended, we cannot include any more patients in it, but we plan to start another study addressing unsolved problems in oral anticoagulation in AF patients (including patients with LAT/SEC) and we will observe your kind recommendations when elaborating the design of this study.
- To your comments about the high percentage of thrombosis and sludge among the investigated patients:
The high incidence of SEC/LAT could be explained with:
- The possible longer duration of AF episodes (>48 h) despite of recognized paroxysmal type of the arrhythmia at the moment of the diagnosis;
- Lack of prior/ongoing anticoagulant therapy;
- High-risk left atrial appendage morphology (especially number of lobes);
The higher prevalence of LAT/SEC in our study was one of the reasons we decided to publish our result.
We completely agree with you that we cannot draw general conclusions over the entire population with AF and LAT/SEC – our conclusions are just hypothesis-generated. Our data suggest only that in patients with paroxysmal AF, considered for cardioversion additional factors such as number of LA lobes, LA blood velocity and LA size should be taken into account before taking decision for the minimal period of anticoagulation, particularly if the patients do not agree to have a control TEE.
Thank you once again for your time and valuable comments, that would definitely help us to improve our future clinical investigations among patients with AF!
Reviewer 2 Report
Author present an interesting study where they have investigated the effect of DOACs in prevention of LAT in AF patients undergoing cardioversion. There are a couple of points author need to revisit:
- It seems that the largest cohort (~75%) in your study was on dabigatran 150 mg bid. Have you tried to analyze dabigatran 150 mg data only to see if the study outcome would sustain? This approach would remove pharmacological variation seen in the study, and could provide further evidence.
- Also, it seems that LAA filling and emptying velocities have inverse correlation with LAT. Authors declare that LAA velocities, etc. are the strongest predictors for persistence of LAT, however, it is widely accepted that this is rather a causation of thrombus formation within LAA due to blood stasis. More discussion is needed in this perspective.
Minor:
- Line 37-39 needs citation.
- Warfarin, dabigatran and others should be written in lower case since it's not brand name.
- It'd be wise to include dosages of DOACs in Figure 1 for better readability.
- "Conclusions" section has to be revised.
Author Response
Dear Reviewer,
Thank you for your time reviewing our manuscript and the valuable comments and recommendations you have made! We observed all recommendations about corrections/additions to our manuscript.
- To your comment “It seems that the largest cohort (~75%) in your study was on dabigatran 150 mg bid. Have you tried to analyze dabigatran 150 mg data only to see if the study outcome would sustain? This approach would remove pharmacological variation seen in the study, and could provide further evidence”:
Our study was not aimed and designed to compare the effects of different DOACs and their dosing regimens. The number of patients on dabigatran 110 mg b.i.d, apixaban & rivaroxaban in our study was too small to compare NOACs to each other. Actually, a sub-analysis of patients on dabigatran 150 mg data had been conducted because the majority of patients had been treated with this NOAC and dose. However, the results did not differ significantly compared to results from the general analysis (all patients included), so we decided to present the results from the analysis of the entire group. We are planning to conduct another study addressing unsolved problems in oral anticoagulation in AF patients, including patients with LAT/SEC. The design of this study will include “inter-DOAC” comparison, moreover we already have all four NOACs (including edoxaban) in Bulgaria available for treatment of AF patients.
2. To your comment “Also, it seems that LAA filling and emptying velocities have inverse correlation with LAT. Authors declare that LAA velocities, etc. are the strongest predictors for persistence of LAT, however, it is widely accepted that this is rather a causation of thrombus formation within LAA due to blood stasis. More discussion is needed in this perspective.”
You are right, the decreased LAA filling and emptying velocities are a causation of thrombus formation within LAA due to blood stasis. However, these velocities are dependent on: (1) The hemodynamic changes in AF per se; (2) Specific LAA morphology and also (3) Presence of LAT. Our data showed an association between some TEE parameters (LA volume index ≥ 34 ml/m2, LAA emptying velocity < 20 cm/s and LAA lobes >2) and the risk of LAT persistence after DOAC treatment.
In terms of the recommended minor changes:
- We added citations to Lines 37-39 (current Lines 39-41 after some other corrections have been made);
- The NOACs - warfarin, dabigatran, rivaroxaban and apixaban were written in lower case;
- We have amended figure 1 as recommended by the Reviewer, adding the dosages of the NOACs for better readability.
- The “Conclusion” section has been revised, both in the abstract and the main text.
Thank you again for your time and valuable comments and recommendations, that improved the quality of our article and would definitely help us to improve our future clinical investigations among patients with AF!
This manuscript is a resubmission of an earlier submission. The following is a list of the peer review reports and author responses from that submission.
Round 1
Reviewer 1 Report
In this study the Authors investigated the prevalence of SEC/LAT in patients with paroxysmal/unknown duration of AF, before DOAC treatment and 3 weeks later, thus evaluating the potential of DOACs to resolve these pathological conditions. The manuscript is well written and structured, the information it is provided is clear.
However, some points have to be cleared:
- The main limitation of the study is the very small sample size. I do not believe that your observation on only 51 patients allows you to conclude anything. I strongly encourage the Authors to include more patients and try to calculate a study sample size that allows you to demonstrate a primary end-point (LAT reduction, SEC reduction, and so on).
- You have enrolled “patients with paroxysmal, non-valvular AF, planned for pharmacological/electrical cardioversion” What does it mean? Why did you perform a cardioversion in a patient with paroxysmal AF?
- On line 206 you state that “Another factor contributing to this prevalence is the possible long duration (much longer than 48 h) of the patients with AF of unknown duration in our study” If you have enrolled patients with paroxysmal AF data on AF duration should be known? Please clarify this crucial point.
- Another big limitation is the very high incidence of SEC/LAT reported in your study population. As, you have assessed in the Discussion, it is over 3 times that reported in several and larger studies. It is very hard to find a clinical significance in your findings.
Reviewer 2 Report
In their article,” Are three weeks of oral anticoagulation sufficient for safe cardioversion in atrial fibrillation?”, Stefan Naydenov et al. set out to study the prevalence of spontaneous echo contrast and left atrial thrombosis before and after 3 weeks of direct oral anticoagulant treatment.
They conclude that minimal period of 3 weeks of oral anticoagulation led to SEC/LAT resolution in 47% of their studied patients. They propose that LA/LAA morphology and function should be taken into account when determining the duration of DOAC treatment before planned cardioversion.
The study certainly explores an important topic. However, there are issues outlined below which first need to be addressed:
Major:
There are multiple studies investigated the hypothesis proposed in this article. Although this is an important topic, these prior studies reduce the novelty and potential impact of the current study. The authors should discuss more regarding the novelty of their findings and what would be the clinical implications of their study. They should also add a section regarding the limitations of their study.
Reviewer 3 Report
Oral anticoagulant (OAC) therapy with vitamin K antagonists (VKAs) and new oral anticoagulants (NOACs) is highly effective in the prevention of stroke for patients with atrial fibrillation (AF). The use of NOACs gave important benefits compared to VKAs in reducing the probability of major hemorrhagic events including intracranial bleeding (IB) with similar or better stroke prevention and easier management.The authors studied a group of patients with atrial fibrillation (AF), lasting >48 h, considered for cardioversion. For these patients are recommended ≥3 weeks of oral anticoagulation before sinus rhythm restoration because of high risk of development of left atrial thrombosis (LAT) and stroke.This is an open-label study, aiming to investigate the prevalence of spontaneous echo contrast (SEC) andLAT before and after 3 weeks of direct oral anticoagulant (DOAC) treatment. The authors included 51 consecutive patients (50.9% males), mean age 69.3±7.4 years with paroxysmal/unknown duration of AF, 14considered for cardioversion, who agreed to have transesophageal echocardiography at enrollment 15 and 3 weeks later.At baseline SEC was present in 26 (50.9%) and LAT in 10 (19.6%) of 51 16 patients. After 3 weeks on DOAC, SEC persisted in 12 (25.0%) and LAT in 7 (14.5%) of 48 patients, 17 p < 0.05 vs baseline. Factors, associated most strongly with persistence of SEC/LAT were left atrial 18 appendage (LAA) emptying velocity < 20 cm/s (OR = 2.82), LAA lobes >2 (OR = 1.84) and indexed 19 left atrial volume ≥ 34 ml/m2 (OR = 1.37).The conclusions are that the recommended minimal period of 3 weeks 20 of oral anticoagulation lead to SEC/LAT resolution in 47% of our patients and LA/LAA 21 morphology and function should be taken into account when determining the duration of DOAC 22 treatment before planned cardioversion.
The study is interesting but presents several critical issues. The major limitations are that the percentage of thrombosis and sludge is very high and the number of patients analyzed is too low to draw conclusions. The same goes for the morphology of the auricle which in some cases is probably at greater embolic risk but also in this case it is necessary to have a larger study population.